# Statistical Hypothesis Testing for Auditing Robustness in Language Models

Paulius Rauba [1]   Qiyao Wei [1]   Mihaela van der Schaar [1]

## Abstract

Consider the problem of testing whether the outputs of a large language model (LLM) system change under an arbitrary intervention, such as an input perturbation or changing the model variant. We cannot simply compare two LLM outputs since they might differ due to the stochastic nature of the system, nor can we compare the entire output distribution due to computational intractability. While existing methods for analyzing text-based outputs exist, they focus on fundamentally different problems, such as measuring bias or fairness. To this end, we introduce distribution-based perturbation analysis, a framework that reformulates LLM perturbation analysis as a frequentist hypothesis testing problem. We construct empirical null and alternative output distributions within a low-dimensional semantic similarity space via Monte Carlo sampling, enabling tractable inference without restrictive distributional assumptions. The framework is (i) model-agnostic, (ii) supports the evaluation of arbitrary input perturbations on any black-box LLM, (iii) yields interpretable p-values; (iv) supports multiple perturbations via controlled error rates; and (v) provides scalar effect sizes. We demonstrate the usefulness of the framework across multiple case studies, showing how we can quantify response changes, measure true/false positive rates, and evaluate alignment with reference models. Above all, we see this as a reliable frequentist hypothesis testing framework for LLM auditing.

## 1. Introduction

Large language models (LLMs) generate outputs conditioned on textual inputs by iteratively sampling from a distribution of tokens. Therefore, the outputs of LLMs exhibit inherent variability due to the stochastic sampling process, a process controlled via parameters such as temperature or top-k (Renze & Guven, 2024). This means that evaluating how interventions—such as input perturbations, model variant changes, prompt system modifications—affect the output, is not straightforward (Romero-Alvarado et al., 2024). Understanding and quantifying the effects of such perturbations is crucial in high-stakes applications—such as legal document drafting or medical diagnosis—where errors or unintended behavior could have significant consequences (Meskó & Topol, 2023; Helberger et al., 2023).

Systematic evaluation of output responses to input perturbation is fundamental to comprehending LLM behavior. It provides quantitative insights into model robustness and output consistency across diverse input conditions. Statistical hypothesis testing of model responses can be used for quantifying these effects and isolating the effects of random variability. There are multiple examples where this is directly relevant. First, it helps with *vulnerability identification* by quantifying potential vulnerabilities to adversarial attacks (Goodfellow et al., 2014). Second, it aids with *bias discovery*, whereby latent biases or unintended behaviors may not become apparent through single-dimension auditing approaches (Ribeiro et al., 2016). Third, it can work within a *compliance framework*. Measurable frameworks for assessing model behavior are essential for compliance with emerging ethical and legal accountability regulations (Doshi-Velez et al., 2017). In the context of language models, subtle changes in patient history could lead to wildly different diagnoses; and patients with nearly identical health records might receive drastically different treatment plans due to minor grammatical changes. Given these critical functions, there is a clear need for a comprehensive LLM auditing framework centered around reliable statistics tools.

Current methods for analyzing LLM behavior changes often focus on simplistic metrics, such as word overlap or direct log-probability comparisons. While effective in certain cases, these approaches fail to account for the nuanced, high-dimensional nature of semantic information processed by LLMs (Bhandari et al., 2020; Lin, 2004). Moreover, existing methods typically lack rigorous statistical foundations, making it difficult to disentangle meaningful changes in model behavior from intrinsic randomness in the output generation process. Efforts to address this have included

---

[1]University of Cambridge. Correspondence to: Paulius Rauba <pr501@cam.ac.uk>.

*Proceedings of the $42^{nd}$ International Conference on Machine Learning*, Vancouver, Canada. PMLR 267, 2025. Copyright 2025 by the author(s).

specialized attribution methods, feature importance techniques, and counterfactual reasoning frameworks (Ribeiro et al., 2016; Garg et al., 2019). However, these approaches are often model-specific, rely on restrictive assumptions about the data or model, or fail to provide interpretable and generalizable metrics.

**Our solution**. In this work, we introduce distribution-based perturbation analysis (DBPA), a framework that reformulates the problem of LLM perturbation analysis as a frequentist hypothesis testing task. DBPA constructs empirical output distributions using Monte Carlo sampling to capture the inherent stochasticity of LLMs, and enables statistical hypothesis testing of perturbation effects within a low-dimensional semantic similarity space. By using statistical hypothesis testing, the framework enables robust, interpretable inferences about whether and how input perturbations meaningfully influence LLM outputs. DBPA is model-agnostic, computationally efficient, and flexible enough to accommodate arbitrary perturbations on any black-box LLM. It also provides interpretable p-values, scalar effect sizes, and supports multiple testing with controlled error rates, making it a versatile tool for post-hoc interpretability and reliability assessments of LLMs.[1]

**Contributions**. ① We identify limitations in existing methods for evaluating language model outputs (Sec. 2). ② We introduce distribution-based perturbation analysis which is a model-agnostic sensitivity technique that can test the effect of *any perturbation* with statistical significance measures (Sec. 3). ③ We perform multiple case studies to show the usefulness of DBPA (Sec. 4).

## 2. An analysis of viewing LLM outputs through frequentist hypothesis testing

### 2.1. Problem formulation

Let $\mathcal{X}$ denote the input space and $\mathcal{Y}$ the output space of a machine learning system. We define the system as a stochastic mapping $\mathcal{S} : \mathcal{X} \rightarrow \mathcal{P}(\mathcal{Y})$, where $\mathcal{P}(\mathcal{Y})$ is the space of probability distributions over $\mathcal{Y}$. This formulation captures the inherent stochasticity of modern ML systems, including LLMs.

Our objective is to address the following research question: Given an ML system $\mathcal{S}$, an input $x \in \mathcal{X}$, and an input perturbation $\Delta_x$, how can we systematically measure and interpret the impact of $\Delta_x$ on the output distribution of $\mathcal{S}(x)$ under a general notion of sensitivity?

**Definition 2.1** (Sensitivity). The sensitivity of a machine learning system $\mathcal{S}$ with respect to an input perturbation $\Delta_x$ :

[1]Code can be found at https://github.com/vanderschaarlab/dbpa

$\mathcal{X} \rightarrow \mathcal{X}$, or a system perturbation $\Delta_\mathcal{S} : \mathcal{S} \rightarrow \mathcal{S}$, at input $x \in \mathcal{X}$, is characterized by a non-negative discrepancy measure $d$ between the output distributions $\mathcal{S}(x)$ and $\mathcal{S}(\Delta_x(x))$, or $\mathcal{S}(x)$ and $\Delta_\mathcal{S}(\mathcal{S})(x)$, respectively.

We aim to perform statistical hypothesis testing on black-box LLMs $\mathcal{S}$ with respect to the system and input. Therefore, (i) *we do not assume access to the underlying system's parameters*, such as their weights or biases; and (ii) *we make no assumptions about access to ground-truth labels*, as the evaluation is done with respect to the generated outputs of the system.

### 2.2. Distribution testing as a frequentist hypothesis-testing problem

The challenge of assessing the sensitivity of $\mathcal{S}$ can be reframed as a frequentist hypothesis testing problem. A common approach involves querying a language model once to obtain an answer, modifying the input, querying again, and comparing the outputs:

$$y \sim \mathcal{S}(x), \quad y' \sim \mathcal{S}(x'). \tag{1}$$

This approach suffers from high variance and therefore low statistical efficiency due to the stochastic nature of LLM systems. Since $\mathcal{S}(x)$ generates a random variable for any fixed input $x$, both $y$ and $y'$ represent single realizations from a distribution of outputs. Any observed difference between $y$ and $y'$ could therefore arise from inherent randomness rather than a true effect of $\Delta_x$.

To address this limitation, we propose reframing the problem from the lens of *distribution testing*. Instead of comparing individual outputs, we would ideally like to compare the entire output distributions. For notational convenience, let $\mathcal{D}_x := \mathcal{S}(x)$ denote the output distribution for input $x$. Let $Y_x$ and $Y_{x'}$ be random variables where $Y_x \sim \mathcal{D}_x$ and $Y_{x'} \sim \mathcal{D}_{x'}$.

**Definition 2.2** (Output Distribution). For a given input $x \in \mathcal{X}$ and an LLM $\mathcal{S} : \mathcal{X} \rightarrow \mathcal{P}(\mathcal{Y})$, the output distribution $\mathcal{D}_x$ is the probability distribution from which the random variable $Y_x$ is drawn when generated by $\mathcal{S}(x)$. That is, $Y_x \sim \mathcal{D}_x$. The probability mass function is given by:

$$p_x(y) = \mathbb{P}(Y_x = y), \quad \forall y \in \mathcal{Y}. \tag{2}$$

Our goal is to determine whether these distributions differ significantly. This can be formulated as a hypothesis test:

$$H_0 : \mathcal{D}_x = \mathcal{D}_{x'} \quad \text{(The perturbation has no effect)} \tag{3}$$
$$H_1 : \mathcal{D}_x \neq \mathcal{D}_{x'} \quad \text{(The perturbation has an effect)} \tag{4}$$

Here, $x' := \Delta_x(x)$ is shorthand for the perturbed input. The primary benefit of such a distributional formulation is that it

captures the full stochastic behavior of $\mathcal{S}$ instead of just a single realization. This means we could perform statistical inference by directly comparing these distributions and understanding how much the outputs have shifted across the whole output space, such detecting subtle shifts that might not be apparent from individual samples.

> **Takeaway**. To understand the sensitivity of a language model to changes in input, we need to look at the entire range of possible outputs it can produce, not just single examples.

Clearly, there are challenges in directly using this framework in the context of language models: how do we *practically* estimate $\mathcal{D}_x$ or how to practically interpret it? We discuss these questions next.

### 2.3. Challenges with analyzing output distributions

There are two primary challenges in comparing output distributions to evaluate the effect of an input perturbation on the output: computational intractability and poor interpretability.

▶ **Challenge 1: Computational intractability**. Even when we restrict the language model to sequences of a fixed length $L$, the output space is exponential: $\mathcal{Y} = V^L$ with $|\mathcal{Y}| = |V|^L$. For a given input $x$, the distribution $\mathcal{D}_x$ is fully described by its probability mass function

$$p_x(\mathbf{y}) = \prod_{t=1}^{L} p\big(y_t \mid y_{<t}, x\big), \quad \mathbf{y} = (y_1, \ldots, y_L) \in \mathcal{Y}. \tag{5}$$

Enumerating $p_x(\mathbf{y})$ for every $\mathbf{y} \in \mathcal{Y}$ requires $O(|V|^L)$ evaluations and is therefore infeasible in practice.

▶ **Challenge 2: Interpretability**. A second major issue is that the distribution $\mathcal{D}_x$ does not provide an interpretable understanding of the LLM output. As LLMs are increasingly being employed as reasoning engines (Yao et al., 2022; Hao et al., 2023), we care about whether their outputs differ semantically, not probabilistically. For instance, suppose $\mathcal{S}$ outputs two answers to a question on treatment recommendations

$y_1$: "*Targeted radiation therapy is suggested*", $y_2$: "*We suggest targeted radiation therapy*".

The two outputs may have different probabilities under $p_x$ even though, semantically, they convey identical recommendations.

Ideally, we would like to be able to resolve both issues at the same time — (i) be able to computationally approximate the distribution *and* (ii) evaluate whether the differences are semantically meaningful, not just probabilistically dif-

ferent. We show how to achieve both with finite-sample approximations.

> **Takeaway**. Analyzing output distributions of language models faces two unique challenges: the computational intractability due to the enormous output space, and the need for semantic rather than just probabilistic interpretation of differences.

### 2.4. Finite sample approximation to output distributions

We have established that using distributions to analyze LLM outputs offers significant advantages compared to using a single output. These advantages come with two practical challenges: (i) computational intractability and (ii) poor interpretability. Here, we suggest that using finite sample approximations can resolve both challenges.

▶ **Addressing challenge 1: Computational complexity**. We address the computational complexity by Monte Carlo sampling. We define the finite sample approximations of the output distributions for an input $x \in \mathcal{X}$ and its perturbation $x'$ as:

$$\hat{\mathcal{D}}_x = \{y_i\}_{i=1}^{k}, \quad y_i \overset{i.i.d.}{\sim} \mathcal{S}(x), \tag{6}$$

$$\hat{\mathcal{D}}_{x'} = \{y_i'\}_{i=1}^{k}, \quad y_i' \overset{i.i.d.}{\sim} \mathcal{S}(x') \tag{7}$$

where $k$ is the sample size. The choice of $k$ affects both the variance of our estimator (which scales as $O(1/k)$) and the power of subsequent hypothesis tests. The number of MC samples $k$ can be treated as a hyperparameter and it should be adapted based on the problem setup (different needs for similarity variance and statistical power).

▶ **Addressing challenge 2: Interpretability**. Given a finite set of samples, we would like to measure how much the output varies given an input perturbation. To quantify the distributional changes induced by input perturbations, we introduce the similarity measure $s : \mathcal{Y} \times \mathcal{Y} \to \mathbb{R}$. This measure allows us to construct empirical distributions of pairwise similarities:

$$P_0 = \{s(y_i, y_j) : y_i, y_j \in \hat{\mathcal{D}}_x, i \neq j\}, \tag{8}$$

$$P_1 = \{s(y_i, y_j') : y_i \in \hat{\mathcal{D}}_x, y_j' \in \hat{\mathcal{D}}_{x'}\} \tag{9}$$

Here, $P_0$ captures the intrinsic variability within the original output distribution, whereas $P_1$ captures the cross-distribution similarities between the original and perturbed outputs. We have therefore constructed two similarity distributions which represent the variability in answer similarities as a proxy for sensitivity (Definition 2.1).

> **Takeaway**. Finite sample approximations using Monte Carlo sampling and pairwise similarity measures offer a computationally feasible and interpretable approach to analyzing output distributions of language models.

## 3. Distribution-based perturbation analysis

We present a model-agnostic methodology for assessing the sensitivity of LLMs to perturbations. Our approach avoids restrictive distributional assumptions and utilizes the entire output distribution of $\mathcal{S}$, capturing the intrinsic variability in LLM responses. We enable frequentist statistical hypothesis testing using p-values through the construction of null and alternative distributions. Importantly, our framework is applicable to *any perturbation* and *any language model*, with the minimal requirement of being able to sample from the language model's output distribution and construct embeddings. Henceforth, we assume that the embedding function is deterministic and stable, while the embeddings are *semantic-preserving*: if two outputs are judged equally similar by humans, the distance between their embeddings should be equal up to monotone transform.

### 3.1. The procedure

**Objective**. With the approach, we are able to evaluate two values. First, we calculate the *effect size* ("by how much has the response distribution shifted"?). Second, we calculate the *p-value* ("is the difference between the new and old distributions significant"?). We follow a simple procedure, the general form of which is outlined below.

---

**Distribution-based perturbation analysis: a quick overview of the procedure**

---

Distribution-based perturbation analysis proceeds in four steps: response sampling, distribution construction, distributional comparison, and statistical inference.

**I. Response Sampling**. Draw $k$ independent outputs from the original prompt and $k$ from the perturbed prompt

$$\hat{\mathcal{D}}_x = \{y_i\}_{i=1}^k, \qquad \hat{\mathcal{D}}_{x'} = \{y_i'\}_{i=1}^k,$$

where $y_i \overset{i.i.d.}{\sim} \mathcal{S}(x)$ and $y_i' \overset{i.i.d.}{\sim} \mathcal{S}(x')$ with $x' := \Delta_x(x)$. Define the pooled vector $Z = (z_1, \ldots, z_{2k})$ with

$$z_i = y_i \quad (1 \le i \le k), \qquad z_{k+i} = y_i' \quad (1 \le i \le k).$$

**II. Distribution construction**. Using a similarity function $s : \mathcal{Y} \times \mathcal{Y} \to \mathbb{R}$, build

$$P_0 = \{ s(y_i, y_j) : 1 \le i < j \le k \},$$
$$P_1 = \{ s(y_i, y_j') : 1 \le i, j \le k \}.$$

**III. Distributional comparison**. Measure the discrepancy between $P_0$ and $P_1$ with any non-negative functional

$$\omega : \mathcal{P} \times \mathcal{P} \longrightarrow \mathbb{R}_{\ge 0}, \qquad T_{\text{obs}} = \omega(P_0, P_1).$$

**IV. Statistical inference**. Formulate the hypotheses

$$H_0 : \mathcal{S}(x) = \mathcal{S}(x'), \qquad H_1 : \mathcal{S}(x) \ne \mathcal{S}(x').$$

We can evaluate this hypothesis via a simple permutation procedure described in Algorithm 1.
**Objective**. If $\hat{p}$ is small, this suggests that $T_{\text{obs}}$ is unusually large relative to its null distribution. The value $T_{\text{obs}}$ itself serves as the effect-size estimate, whereas the permutation test provides frequentist p-values (Knijnenburg et al., 2009; Phipson & Smyth, 2010).

---

We can then perform statistical inference by permutation testing. The procedure is simple and requires only minimal exchangeability assumptions on raw outputs. Exchangeability would not be guaranteed by permuting the similarity scores because they are correlated.

**Assumption 3.1** (Exchangeability Under the Null Hypothesis). Under the null hypothesis $H_0 : \mathcal{D}_x = \mathcal{D}_{x'}$, the pooled outputs $Z = (y_1, \ldots, y_k, y_1', \ldots, y_k')$ are exchangeable. That is, for every permutation $\pi$ of the index set $\{1, \ldots, 2k\}$, we have

$$(y_1, \ldots, y_k, y_1', \ldots, y_k') \overset{d}{=} (y_{\pi(1)}, \ldots, y_{\pi(2k)}).$$

With this assumption, we can formalize the algorithm for permutation testing as follows.

Under the strict null hypothesis $H_0 : \mathcal{D}_x = \mathcal{D}_{x'}$, any function of the pooled outputs is exchangeable, so our permutation-based p-value is valid. We choose to summarize outputs only via a semantic similarity function $s$ (e.g., cosine on sentence embeddings). Thus, the test is sensitive only to differences that appear in those similarity scores. Rejection implies a detectable shift in meaning; non-rejection means "no detectable semantic shift under $s$," but does *not* imply that the full text-generation distribution is identical. We discuss why this is desirable in *Challenge 2* of Sec. 2.3.

**Algorithm 1** Permutation Testing for Distribution-based Perturbation Analysis

---

**Require:** Pooled vector $Z = (z_1, \ldots, z_{2k})$, similarity function $s$, discrepancy measure $\omega$, number of permutations $B$

**Ensure:** $p$-value $\hat{p}$

1: Compute observed test statistic:
2:     $P_0 = \{s(z_i, z_j) : 1 \leq i < j \leq k\}$
3:     $P_1 = \{s(z_i, z_j) : 1 \leq i \leq k < j \leq 2k\}$
4:     $T_{\text{obs}} = \omega(P_0, P_1)$
5: Initialize counter: $\text{count} = 0$
6: **for** $\pi = 1$ to $B$ **do**
7:     Randomly select subset $I \subset \{1, \ldots, 2k\}$ of size $k$
8:     $O = \{z_i : i \in I\}, \ P = \{z_j : j \notin I\}$
9:     $P_0^{(\pi)} = \{s(u, v) : u, v \in O, u \neq v\}$
10:    $P_1^{(\pi)} = \{s(u, v) : u \in O, v \in P\}$
11:    $T^{(\pi)} = \omega(P_0^{(\pi)}, P_1^{(\pi)})$
12:    **if** $T^{(\pi)} \geq T_{\text{obs}}$ **then**
13:        $\text{count} = \text{count} + 1$
14:    **end if**
15: **end for**
16: $\hat{p} = \frac{1 + \text{count}}{1 + B}$
     **return** $\hat{p}$

---

### 3.2. From theory to practice: design choices to consider for distribution-based perturbation analysis

There are a few practical implementation essentials to take into account for the outlined procedure.

► **Why use *scalar* pairwise cosine similarities instead of directly using high-dimensional embeddings?** Employing unreduced embeddings to construct null and alternative distributions faces two main obstacles. *(a) High dimensionality.* Embedding spaces typically have hundreds or thousands of dimensions, so estimating a full distribution in that space is intractable because of the curse of dimensionality. *(b) Excess semantic information.* Embeddings capture rich semantics—most of which are irrelevant when the sole aim is to quantify *how much* the responses move as the input is perturbed, rather than *where* each response lives in the embedding manifold. Reducing each *pair* of embeddings to a single similarity score sidesteps both issues.

► **What is an appropriate similarity function?** Instead of an explicit dimensionality-reduction map, we apply a similarity function that takes two outputs to $\mathbb{R}$. Among several viable choices (e.g., negative $\ell_1$ or $\ell_2$ distance), we adopt the cosine similarity

$$s(y_i, y_j) = \frac{\langle e(y_i), e(y_j) \rangle}{\|e(y_i)\| \, \|e(y_j)\|},$$

where $e(\cdot)$ is an assumed deterministic, stable embedding function (*ada-002* for most experiments). Cosine similarity

is not a metric because the triangle inequality fails (Schubert, 2021). However, we only require a meaning-preserving, bounded similarity, not a metric structure.

► **What is an appropriate distance measure $\omega$?** The reason why there exists a choice for $\omega$ is that we are dealing with the comparison between two distributions. This is different from traditional resampling-based approaches that construct a null distribution and evaluate a single instance against it (Yu, 2003). While the choice for $\omega$ might vary depending on the application, we employ the Jensen-Shannon divergence (JSD) as a measure for $\omega$: $\text{JSD}(P_0 \| P_1) = \frac{1}{2}\left(D_{\text{KL}}(P_0 \| M) + D_{\text{KL}}(P_1 \| M)\right)$, where $M = \frac{1}{2}(P_0 + P_1)$ and $D_{\text{KL}}$ is the Kullback-Leibler divergence. We convert each multiset into an empirical probability measure and the JSD is taken between those empirical distributions. This is because JSD has three useful properties for evaluating distributions: (i) symmetry, ensuring that the measure is invariant to the order of the distributions being compared; (ii) boundedness, providing a consistent scale for interpretation across different inputs and perturbations; and (iii) sensitivity to differences in both the location and shape of the distributions.

► **Why perform permutation-based testing instead of direct Monte-Carlo sampling from $\mathcal{S}$?** In principle, one could approximate null distributions by repeatedly sampling $\mathcal{S}(x)$ and $\mathcal{S}(\Delta_x(x))$. However, new model calls are typically the dominant cost. A permutation test reuses the same $2k$ generated outputs, yet—because the pooled vector is exchangeable under $H_0$ (Assumption 3.1)—delivers a valid $p$-value with far fewer queries. We expand on this in Appendix A.

► **How should we control multiplicity when testing many perturbations?** When DBPA is applied to a *family* of perturbations $\{\Delta_i\}_{i=1}^m$, each hypothesis

$$H_0^{(i)} : \mathcal{S}(x) = \mathcal{S}(\Delta_i(x))$$

is assessed with its own permutation $p$-value $\hat{p}_i$. Running $m$ independent tests in parallel inflates the probability of at least one Type-I error. To keep the family-wise error rate (FWER) at a desired level $\alpha$, one can default to the *Bonferroni correction*:

$$\text{FWER} = 1 - (1 - \alpha)^m \approx m\alpha, \quad \alpha_{\text{adj}} = \frac{\alpha}{m}, \ p_{\text{crit}} = \frac{\alpha}{m}.$$

A common countermeasure is to increase the Monte Carlo sample size $k$ used to build the empirical distributions $P_0$ and $P_1$. Under the usual normal approximation,

$$k_{\text{adj}} \ \propto \ \left(\frac{z_{1-\alpha/(2m)} + z_{1-\beta}}{\omega}\right)^2,$$

where $\omega$ denotes the target effect size, $z_q$ is the $q$-th standard-normal quantile, and $\beta$ the tolerated Type-II error. Hence,

| Experiment | Purpose | Finding | Name | Case study | Insight |
|---|---|---|---|---|---|
| Persona Perturbation | Test role instruction effect | Large LMs stable on medical roles; small LMs variable | Table 1 | ✓ | ✗ |
| TPR/FPR Trade-off | Measure TPR/FPR in healthcare | Model choice depends on FPR threshold | Fig. 2 | ✓ | ✗ |
| Model Alignment | Compare outputs to GPT-4 | Most models align; some small LMs differ | Fig. 3 | ✓ | ✗ |
| Input Length Impact | Assess effect of prompt length | Longer contexts reduce detectability | Table A.1 | ✗ | ✓ |
| Alt. Distance Measures | Compare JSD, Euclid., etc. | Significance decisions consistent across measures | Table A.2 | ✗ | ✓ |
| Embedding Costs | Estimate embedding expenses | Perturbations remain inexpensive even at scale | Table A.3 | ✗ | ✓ |
| Text Similarity Alt. | Use BLEU/ROUGE in DBPA | Highly sensitive to small changes; less reliable | Table A.4 | ✗ | ✓ |
| Impact of Embeddings | Test various embedding models | Creative personas' effects consistent across embeds | Table A.5 | ✗ | ✓ |

Table 1: **Bite-sized summary of experiments and case studies conducted**. We conduct three case studies (main section) and five additional insight studies (appendix) to understand why, how, and when our framework might be useful.

practitioners can trade additional model calls for restored power while still enjoying rigorous FWER control; less conservative multiplicity-adjustment methods may be substituted when appropriate.

## 4. Case studies

We demonstrate the effectiveness of our method on a variety of use cases. In the following subsections, we will show that our method can (1) capture those answer divergences that are significant and those that are not under perturbation (2) analyze the robustness of language models to irrelevant changes in the prompt (3) evaluate alignment with reference language model. By default, we run the experiment over 5 seeds, and report the mean and standard deviation of the measurements. We calculate the distance measure $\omega$, computed as the JSD distance between the null and alternative distributions, and the p-values.

### 4.1. Case study 1: How much do responses change under different input perturbations?

Language models are known to have strong role-playing abilities which shape their responses (Wang et al., 2023; Kong et al., 2023; Chen et al., 2024). We use this property to evaluate response changes under different assigned roles.

• **Setup**. We evaluate whether we can capture LLM response variability to prompt perturbations. We compare response distributions before and after a prompt perturbation. First, we query an LLM with a healthcare question to establish a null distribution from the baseline responses. Then we pre-pend a role-playing instruction of the form '*Act as {role}*', where *{role}* describes a specific given role. We divide the setups into medical professions (medical supervisor, therapist, doctor, medical student) and other roles (comedian, robot from the future, neurips reviewer, child). We then quantify the effect size $\omega$ and the p-value. We run the case study with larger models (GPT-4 and GPT-3.5) as well as smaller open-source models which are generally considered less capable.

• **Goal**. We aim to showcase that we can quantify the shift in the output distribution by pre-pending a different role to a language model and that this shift is both *model* and *prompt* dependent.

The results for *Case Study 1* are presented in Table 2. Given that the scenario in the prompt is a medical one, we would expect that there should be no difference in the similarities of the output distributions for the first four personas (which represent medical professions); and there should be such differences for the other four personas. We see that this is true in larger models (*GPT-4* and *GPT-3.5*) where none of the effect sizes are significant for medical personas and most of them are for the non-medical personas.

On the other hand, smaller models indicate no such relationship between persona and output distribution shift. We interpret this as smaller models showing higher level of answer variability and being generally less consistent / sensitive to specific instructions than larger, more capable models. In fact, we view this analysis as showcasing why using smaller models would be unsuitable because the kinds of responses obtained by adding a persona diverge too much from what we would expect. Changing the number of Monte Carlo samples is one way to make the estimates more precise.

These results directly relate to our primary question of auditing robustness. We observe that smaller models exhibit much greater variability around personas which, intuitively, should not affect the output responses. Therefore, we suggest that smaller models are less robust than larger models.

### 4.2. Case study 2: Can we measure true positive and false positive response changes?

**Measuring robustness of language models by quantifying TPR and FPR**. We adapt our framework to design a new sensitivity measure for language models by balancing the true positive and false positive rates of LLM responses. Let $\{\Delta_i\}_{i=1}^{m}$ be a collection of perturbations, partitioned into control indices $C$ (where outputs should *not* change under a perturbation) and target indices $T$ (where outputs *should* change under a perturbation). For each $i$, define $x_i' = \Delta_i(x)$ and let $p_i$ be the permutation-test p-value comparing $\mathcal{D}_x$

Table 2: Effect-size estimates $\omega$ by persona for all models based on the situation described in Sec. 4.1. Stars denote two-sided significance levels: $^*p < 0.05$, $^{**}p < 0.01$, $^{***}p < 0.001$ calculated using Alg. 1. Higher effect sizes show that the output distribution has shifted given the selected number of Monte Carlo samples. As a helpful proxy, we expect outputs to remain stable under the four medical personas (top of table) but shift under the other four personas (bottom).

| | **Large models** | | **Smaller models** | | | | | | |
| Persona | GPT-4 | GPT-3.5 | LLama-3.1-8B | GPT-2 | Gemma-2-9B | SmolLM-135M | Phi-3-mini | MagicPrompt | Mistral-7B |
|---|---|---|---|---|---|---|---|---|---|
| Doctor | 0.22 | 0.21 | 0.20 | 0.20 | 0.41$^{**}$ | 0.14 | 0.16 | 0.17 | 0.31$^*$ |
| Nurse | 0.20 | 0.20 | 0.25 | 0.30$^*$ | 0.24$^*$ | 0.13 | 0.22 | 0.13 | 0.24 |
| Medical Practitioner | 0.25 | 0.19 | 0.25 | 0.33$^{**}$ | 0.30$^*$ | 0.29$^{**}$ | 0.18 | 0.20 | 0.20 |
| Medical Supervisor | 0.22 | 0.17 | 0.20 | 0.19 | 0.17 | 0.17 | 0.25 | 0.26 | 0.20 |
| Comedian | 0.32$^{***}$ | 0.24 | 0.36$^{***}$ | 0.26 | 0.23 | 0.12 | 0.23 | 0.24 | 0.20 |
| Robot From The Future | 0.15 | 0.26$^*$ | 0.31$^{**}$ | 0.23 | 0.18 | 0.16 | 0.23 | 0.25 | 0.24 |
| NeurIPS Reviewer | 0.28$^{**}$ | 0.28$^*$ | 0.30$^{**}$ | 0.17 | 0.19 | 0.13 | 0.24 | 0.17 | 0.28 |
| Child | 0.27 | 0.33$^{**}$ | 0.19 | 0.27$^*$ | 0.14 | 0.28$^{**}$ | 0.20 | 0.19 | 0.37$^{**}$ |

and $\mathcal{D}_{x'_i}$. Fix a significance level $\alpha$. We declare "change detected" for perturbation $i$ if $p_i < \alpha$. Then FPR $:= \frac{1}{|C|} \sum_{i \in C} \mathbb{I}[p_i < \alpha]$ and TPR $:= \frac{1}{|T|} \sum_{i \in T} \mathbb{I}[p_i < \alpha]$.

Clearly, this can be extended by varying $\alpha \in [0, 1]$. This, first, allows the user to trace the trade-off between FPR and TPR for a given desired error level. Second, it enables to obtain a *global* view of a model's sensitivity to FPR and TPR by summarizing the performance in a single scalar. After obtaining the dataset $(\text{FPR}(\alpha), \text{TPR}(\alpha))$ for $\alpha \in [0, 1]$, the LLM overall trade-off is given by the simple integral

$$\text{AUC} = \int_0^1 \text{TPR}(\alpha)d[\text{FPR}(\alpha)].$$

For our purposes, a higher AUC suggests the LLM can better distinguish target perturbations from control perturbations *within the user-defined setting* of what is target and control. In practice, we compute it with the standard trapezoidal sum over threshold levels. Now, we empirically demonstrate how to use this for evaluating LLMs.

- **Setup**. We create healthcare prompts with patient varying patient features sampled from a distribution and query the LLMs for assessing the cardiovascular disease (CVD) recommendations for the given patient based on the NICE guidelines. We create two types of perturbations: (1) control perturbations that should not change medical recommendations (e.g. changing the patient's name or modifying a covariate that should not affect the guidelines) and (2) target perturbations that *should* change recommendations (e.g. changing a patient's cholesterol or diabetes information to move them into a different risk level which requires a different recommendation). For each perturbation type, we calculate the p-values and obtain TPR and FPR rates.

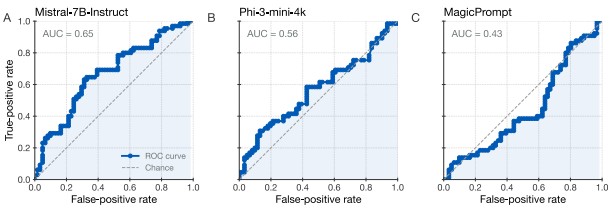

*Higher curves (upper left) indicate better discriminatory power

Figure 1: **TPR/FPR trade-offs for three selected language models**. The figure shows three panels: A, B, and C, describing the relationship between TPR and FPR for three models together with the AUC score evaluated based on the methodology in Sec. 4.2. The ROC-AUC describes the changing performance of the model as we vary $\alpha \in [0, 1]$. Models with higher AUC are better because they can detect true changes and fail to answer differently with irrelevant changes.

- **Goal**. We aim to show that our method can be used to assess LLM reliability for perturbations by quantifying their true positive and false positive rates for custom-designed scenarios.

We run our TPR/FPR analysis and present three models in Fig. 1. We find that not all models are equally strong for the selected cases. For our example, we have demonstrated the best, middle, and worse performing models. A person wanting the highest ROC-AUC over TPR/FPR should choose the model in Panel A. However, we find that while this is an intuitive way to choose a single model, the best model might depend on the exact error rate. For instance, in Fig. 2, we show that in the same example, for an allowed FPR of 1%, we would choose *Phi-3* whereas an allowed FPR of 5% would suggest *Mistral-7B*.

Evidently, there is nothing special about the chosen model or

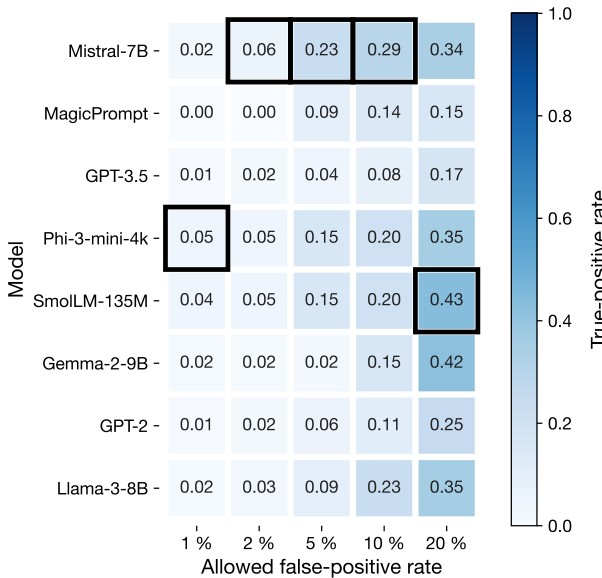

Figure 2: **TPR by selected FPR for multiple models**. We show how we can select the best model with the highest TPR for a selected FPR. The best model for an allowed false positive rate is highlighted in a black box. We find that *the best model to choose might depend on the allowed false-positive rate*. Higher TPR for a given FPR is better.

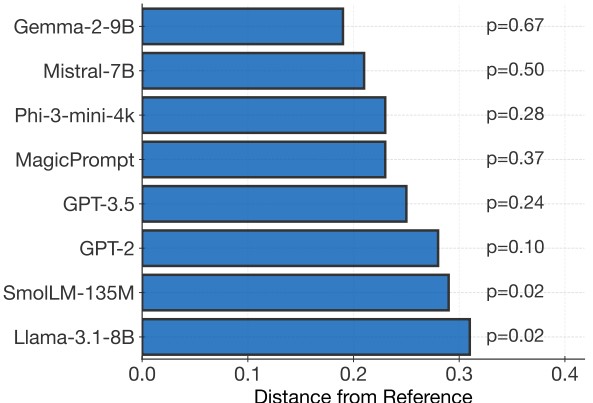

Figure 3: **Distance of responses from a reference language model**. We collect $k$ responses from a base model (GPT-4) and evaluate the responses from multiple other language models reported on the left, compute their $\omega$ and p-values using Alg. 1. Higher distance implies greater differences in similarity distributions.

setup. Such analyses can be performed within any selected language model and any setup. The important requirement for the end user is to specify when a language model should (and should not) have their responses changed. Because the results depend on both the model and the problem setup, we do not find any meaningful interpretation in the AUC-score alone and recommend it merely as a comparative system while evaluating multiple models or systems.

### 4.3. Case study 3: Can we evaluate alignment with a reference language model

Thus far, we have changed the input prompt while holding the model constant. We can also do the opposite—hold our prompt constant while we change the model. This coincides with two formulations in Sec. 2.1. We can use this as a way to evaluate *alignment between different language models*. The results are presented in Fig. 3.

- **Setup**. We evaluate alignment between two language models by sampling responses from a given language model and fixing the input prompt but swapping out the model. Then, the differences between the output distributions correspond to the distance between model responses. We evaluate this across multiple open- and closed-source models.

- **Goal**. We aim to show that we can understand how well two language models answer across their response distributions.

The results in Fig. 3 suggest that we can quantify differences in response distributions between models, providing a concrete measure of inter-model alignment that could be useful for comparative analysis of language model outputs. Given the number of Monte Carlo samples used, only two models—*SmolLM* and *LLama-3.1*—had significantly different responses with other models showcasing less divergence. Clearly, what is "better" depends on the use case: if we aim to ensure that the outputs do not diverge between language models, we desire a low $\omega$; the opposite is true if we expect outputs to change, such as after iterations of applying reward modeling on a language model.

### 4.4. Other evaluations

The case studies above show what we consider to be the most important use cases of hypothesis testing for auditing robustness in language models. Here, we showcase some other use cases and present some analyses we have conducted. Language models struggle to work with long contexts, a phenomenon dubbed "lost in the middle" (Liu et al., 2023) We investigated the effect of having longer input lengths on our Persona experiment (Sec. C.1) and found that it is more difficult to detect changes due to an input perturbation within longer context windows. We evaluate alternative distance measures (Sec. C.2) and find that the evaluations are sensitive to the chosen distance measure but stay comparatively consistent on the significance of the effect sizes.

Because this framework relies on Monte Carlo sampling as well as embedding models, we evaluate the costs of using various embedding models and find that even at large scales, performing such perturbation analyses remains cost efficient (Sec. C.3). We evaluate our model compared to other traditional alternatives which do not operate in semantic space, such as BLEU or ROUGE metrics (Sec. C.4) and evaluate the effect of using different embedding models (Sec. C.5), generally finding that the results are generally sensitive to the embedding for calculating p-values and effect sizes.

## 5. Related work

There are three primary approaches to quantifying text-based outputs that relate to our approach. They include measuring unintended biases in model outputs, developing counterfactual fairness methods, and text summarization metrics.

▶ **Measuring unintended bias**. Overall, the closest related works are in measuring unintended bias (Borkan et al., 2019; Dixon et al., 2018; Park et al., 2018). Such metrics quantify existing biases between subgroups for models. Broader work in the field can be found in (Friedler et al., 2021; Kleinberg et al., 2016; Menon & Williamson, 2018). However, this requires human annotation, relates *only* to fairness, and assumes the existence of reliable labels across subgroups.

▶ **Counterfactual fairness**. This approach (Garg et al., 2019) examines how predictions would change if sensitive attributes were different. It can compute effect sizes but cannot be applied to black-box models, doesn't allow arbitrary perturbations, and doesn't enable statistical inference. It requires human input for labeling a specific attribute (e.g. toxicity) of an answer and makes assumptions, e.g. that non-toxic examples are less likely to contain asymmetric counterfactuals relative to toxic examples.

▶ **Text summarization metrics**. These metrics (Bhandari et al., 2020; Zhang et al., 2019; Zhao et al., 2019; Lin, 2004) evaluate the quality of text summarization. They can compute effect sizes but are not applicable to black-box models, don't allow arbitrary perturbations, and don't enable statistical inference. They require human input and make certain assumptions. Various metrics like BERTScore, MoverScore, and ROUGE variants measure different aspects of similarity between system and reference summaries.

To better explain how we differ, we compare each area based on five important criteria: (i) whether the method can be applied to any black-box model; (ii) whether any perturbation can be applied and measured; (iii) whether the approach enables statistical inference; (iv) whether the approach allows to compute effect sizes of the change; (v) whether there are any assumptions; (vi) whether humans are required as a part of the input. We show this in Table 8.

## 6. Discussion

With the growing need to evaluate LLM systems, we require statistics-based approaches to understand LLM outputs. We know that we cannot simply ask language models to explain whether their answer would be different under given circumstances, as they might not be truthful (Si et al., 2023). Are there ways of having systematic approaches of evaluating how model responses affect the outputs?

In our paper, we provide one way of achieving this. Specifically, we recognize that, in order to fully analyze language models responses, we *must consider the entire output distribution* of the LLM responses. Therefore, comparing single two point estimates is insufficient. To address this, we introduce distribution-based perturbation analysis as a method to quantify the discrepancy between two outputs using Monte Carlo sampling and frequentist-based inference. Importantly, our model has desirable statistical properties that make it directly useful in many application areas, such as being model-agnostic, supporting any input perturbation that the user can specify, providing statistical estimates, providing controlling error rates for multiple perturbation testing, and enabling to compare how different perturbations relate to each other via effect sizes.

We see this being directly useful in high-stakes areas, such as model auditing, model transparency, regulatory compliance, or in domains where language models require interpretability and are used as parts of broader statistical systems. Furthermore, a core component of the hypothesis testing framework is the ability to select and quantify true and false positive rates for a given user problem setup. This gives incredible flexibility to select the best model for a desired false positive rate, a setup extremely useful in high-stakes settings.

**Limitations.** This paper presents the first approach to employing frequentist hypothesis testing to auditing language models. However, the nature of this paper is conceptual in its scope. Future work should add stronger empirical evidence and test the method in other domains. Key design choices—such as distance metrics, embedding functions, and their effect on generalization—would benefit from deeper analysis and clearer selection guidelines that should be established in future work. Additionally, although DBPA offers valuable insights for model auditing, translating these findings into practical strategies for enhancing model robustness and aligning outputs with human preferences remains a significant challenge.

The need for interpretable and statistics-grounded systems to explain black-box behavior and provide guarantees is growing. We hope our work provides a first step in this direction and can be built upon for theoretical and practical use.

# Impact Statement

This paper presents work whose goal is to advance the field of Machine Learning in the field of interpretability. We recast a broad range of evaluation questions as formal hypothesis tests and therefore supply practitioners, policymakers, and independent auditors with a statistically principled tool for detecting genuine behavioural shifts in large language models. We recognize that this might have many great positive implications on safety and auditing of language model systems. At the same time, we acknowledge the dual-use risks, such as adversaries using the logic in the procedure to evade detection or optimize adversarial attacks. However, we think such risks are extremely unlikely in practice.

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

## A. MC-Sampling

To understand why we employ monte-carlo sampling, let's consider the expectation of $\omega$ under both approaches. For permutation testing, this yields:

$$\mathbb{E}[\omega(P_0^*, P_1^*)] = \mathbb{E}[\omega(\{s(z_i, z_j)\}, \{s(z_i, z_k)\})] \quad z_i, z_j, z_j \overset{i.i.d.}{\sim} \mathcal{Z} \tag{10}$$

where $\mathcal{Z} = \{y_i, \ldots, y_k, y_1', \ldots, y_k'\}$ is the combined set of 2k samples from $S(x)$ and $S(\Delta(x))$. For resampling, the expectation is:

$$\mathbb{E}[\omega(P_0', P_1')] = \mathbb{E}[\omega(\{s(y_i, y_j)\}, \{(y_i, y_k')\})] \quad y_i, y_j \overset{i.i.d.}{\sim} S(x), y_k' \overset{i.i.d.}{\sim} S(\Delta(x)) \tag{11}$$

Under $H_0$, $S(x) = S(\Delta(x))$, so all samples in $\mathcal{Z}$ follow the same distribution. Therefore, randomly sampling from $\mathcal{Z}$ is equivalent to sampling from either $S(x)$ or $S(\Delta(x))$. Therefore, the expectations are identical, and the permutation testing will produce the same distribution of $\omega$ values in the limit.

## B. Examples of prompt inputs

### B.1. Example inputs for treatment recommendations

In the exercise for treatment recommendations, we use the following input prompts (where characteristics can vary):

John has these features: Age: 70, BMI: 30, Blood Pressure: 114/61 mmHg, Total Cholesterol: 6.7 mmol/L, HDL Cholesterol: 1.0 mmol/L, Smoking Status: current smoker, Diabetes: no diabetes, Family History of CVD: yes, Ethnicity: Chinese. Based on these characteristics alone, provide recommendations on CVD guidelines based on NICE for this person

John has these features: Age: 70, BMI: 29, Blood Pressure: 114/61 mmHg, Total Cholesterol: 6.7 mmol/L, HDL Cholesterol: 1.0 mmol/L, Smoking Status: current smoker, Diabetes: no diabetes, Family History of CVD: yes, Ethnicity: Chinese. Based on these characteristics alone, provide recommendations on CVD guidelines based on NICE for this person

### B.2. Example outputs for treatment recommendations

The following are example outputs:

```
1    ["Based on these characteristics, NICE guidelines for CVD recommend the following:\n\
     n1. Encouraging John to quit smoking and providing support and resources to help him
     do so.\n2. Promoting a healthy diet and regular physical activity to help lower his
     BMI and improve his cholesterol levels.\n3. Monitoring and managing his blood pressure
      to keep it within a healthy range.\n4. Considering his family history of CVD, it may
     be advisable to assess his overall cardiovascular risk and consider interventions to
     reduce that risk.\n5. Providing culturally sensitive and tailored support for his
     Chinese ethnicity, taking into account any specific risk factors or considerations.\n\
     nIt's important for John to work with his healthcare provider to develop a
     personalized plan for cardiovascular health and to address any individualized risk
     factors or concerns.",
2    "Based on the provided characteristics, John's CVD risk would be assessed as moderate to
      high. The NICE guidelines recommend the following for individuals with similar
     characteristics:\n\n1. Lifestyle changes: John should be advised to make lifestyle
     changes such as quitting smoking, following a healthy diet to reduce cholesterol
     levels and maintain a healthy weight, and engage in regular physical activity.\n\n2.
     Assess CVD risk: John should undergo a formal assessment of CVD risk using a
     cardiovascular risk assessment tool, taking into account his age, BMI, blood pressure,
      cholesterol levels, smoking status, and family history of CVD to determine the need
     for medication or further interventions.\n\n3. Regular monitoring: John should have
     regular monitoring of his blood pressure, cholesterol levels, and overall
     cardiovascular health to manage and reduce his risk of CVD.\n\n4. Consider medication:
      Depending on the results of his CVD risk assessment, John may need to be considered
     for medication such as statins to lower his cholesterol levels and reduce his overall
     CVD risk.\n\nIt's important for John to work closely with a healthcare professional to
      develop a personalized plan for managing his CVD risk based on the NICE guidelines.",
```

3    "Based on the provided characteristics, NICE guidelines for CVD prevention would
     recommend the following:\n\n1. Encouraging John to quit smoking and providing support
     and resources to help him quit.\n2. Recommending lifestyle modifications such as
     regular physical activity and a healthy diet to help manage his BMI and cholesterol
     levels.\n3. Monitoring and managing his blood pressure to keep it within a healthy
     range.\n4. Consideration of cholesterol-lowering medication if lifestyle modifications
      are not sufficient to achieve target cholesterol levels.\n5. Regular monitoring and
     management of other CVD risk factors, especially considering his family history of CVD
     .\n\nIt's important for John to work closely with his healthcare provider to create a
     personalized plan for CVD prevention and management based on his individual risk
     factors and health needs.",

4    "Based on the provided characteristics, the following recommendations can be made for
      CVD guidelines based on NICE:\n\n1. Lifestyle changes: John should be advised to quit
      smoking and adopt a healthier lifestyle, including regular physical activity and a
      balanced diet to help lower his BMI and cholesterol levels.\n\n2. Cardiovascular risk
      assessment: John's overall cardiovascular risk should be assessed using a validated
      tool, taking into account his age, blood pressure, and cholesterol levels.\n\n3.
      Medical assessment: John should be assessed for any potential risk factors for
      cardiovascular disease, such as high blood pressure, and be offered appropriate
      management and treatment as necessary.\n\n4. Family history: Given John's family
      history of CVD, he should be counseled on the potential genetic predisposition to
      cardiovascular disease and the importance of managing other risk factors effectively.\
      n\n5. Ethnicity: Being of Chinese ethnicity, John should be aware that certain
      populations may have a higher risk for specific cardiovascular conditions, and this
      should be taken into consideration when assessing his overall cardiovascular risk.\n\
      nIt is important to note that these recommendations are general and may need to be
      tailored specifically to John's individual health needs and circumstances. It is
      advisable for John to seek personalized medical advice and undergo a comprehensive
      cardiovascular risk assessment and management plan under the care of a healthcare
      professional.",

# C. Extended experiments

## C.1. Impact of prompt length

**Experimental Design.** To address concerns about DBPA's effectiveness on longer and more complex input sequences, we conducted additional experiments varying prompt length from 100 to 900 tokens across different persona instructions. We re-use the same eight distinct personas and vary the input prompt length.

**Main results**. We find that for token counts above 800 tokens, the persona counts do not indicate any significant shifts in the output distribution. The observed pattern of stronger effects in shorter prompts followed by attenuation in longer sequences aligns with well-documented empirical phenomena in large language models, where extended context can lead to more consistent but potentially less distinctive outputs. If anything, this quantitative characterization of prompt length effects could open new ways of looking into context-dependent model behavior and could be another way to provide practitioners with guidance on optimal prompt engineering strategies for different applications.

Table 3: **Effect-size estimates by persona across different token counts.** Stars denote two-sided significance levels: $^*p < 0.05$, $^{**}p < 0.01$, $^{***}p < 0.001$. Higher effect sizes indicate greater distributional shifts in model outputs given the persona instruction. Medical personas (top) are expected to show stable, lower effect sizes, while creative/role-playing personas (bottom) should demonstrate higher variability and stronger effects across token lengths.

| Persona | Token Count | | | | | | | | |
|---|---|---|---|---|---|---|---|---|---|
| | **100** | **200** | **300** | **400** | **500** | **600** | **700** | **800** | **900** |
| Act as a doctor | 0.14 | 0.25 | 0.20 | 0.22 | 0.20 | 0.16 | 0.16 | 0.22 | 0.19 |
| Act as a nurse | 0.18 | 0.27$^{**}$ | 0.23 | 0.20 | 0.25 | 0.37$^{***}$ | 0.19 | 0.17 | 0.18 |
| Act as a medical practitioner | 0.16 | 0.23 | 0.22 | 0.23 | 0.21 | 0.45$^{***}$ | 0.17 | 0.22 | 0.18 |
| Act as a medical supervisor | 0.24 | 0.25 | 0.21 | 0.26 | 0.21 | 0.33$^{**}$ | 0.20 | 0.17 | 0.17 |
| Act as a comedian | 0.24 | 0.41$^{***}$ | 0.34$^{***}$ | 0.26 | 0.41$^{***}$ | 0.18 | 0.28$^*$ | 0.24 | 0.17 |
| Act as a robot from the future | 0.26 | 0.28$^{**}$ | 0.32$^{***}$ | 0.42$^{**}$ | 0.29 | 0.19 | 0.22 | 0.19 | 0.20 |
| Act as a NeurIPS reviewer | 0.40$^{***}$ | 0.36$^{***}$ | 0.39$^{**}$ | 0.35$^*$ | 0.23 | 0.37$^*$ | 0.26 | 0.25 | 0.17 |
| Act as a child | 0.34$^{***}$ | 0.31$^*$ | 0.32$^*$ | 0.27 | 0.37$^{***}$ | 0.26$^*$ | 0.31$^{***}$ | 0.18 | 0.23 |

## C.2. Impact of distance measures

**Experimental Design.** To evaluate the robustness of DBPA across different distributional distance measures, we replicated our core experiments using six measures: Energy distance with cosine similarity, Euclidean distance, Wasserstein distance, Jensen-Shannon divergence (JSD), and Energy distances with L1 and L2 norms.

**Main results.** The results demonstrate that while different distance metrics yield different magnitudes due to their inherent scales and sensitivity properties, the relative patterns across personas remain consistent. Medical personas consistently show lower effect sizes across all metrics, while creative and role-playing personas exhibit substantially higher values. The consistency of persona rankings across metrics validates the robustness of our approach, while the varying sensitivities highlight how different distance measures can provide complementary insights into model behavior under different prompt conditions.

Table 4: Distributional distance measures across personas measuring deviation from baseline outputs. Values represent mean distances with significance levels: $^*p < 0.05$, $^{**}p < 0.01$, $^{***}p < 0.001$. Higher values indicate greater divergence from the model's default response patterns.

| Persona | Distance Measures | | | | | |
| --- | --- | --- | --- | --- | --- | --- |
| | **Energy (Cosine)** | **Euclidean** | **Wasserstein** | **JSD** | **Energy (L1)** | **Energy (L2)** |
| Doctor | 0.01*** | 31.95 | 2.72 | 0.20 | 1.97*** | 0.06*** |
| Nurse | 0.01*** | 34.63 | 2.56 | 0.20 | 1.57*** | 0.05*** |
| Practitioner | 0.01** | 40.88 | 1.57 | 0.22 | 1.27* | 0.04** |
| Supervisor | 0.01*** | 37.34 | 1.60 | 0.21 | 2.15*** | 0.07*** |
| Comedian | 0.11*** | 54.99*** | 4.36*** | 0.31 | 9.11*** | 0.30*** |
| Robot | 0.01*** | 40.95 | 2.47 | 0.22 | 1.71** | 0.05*** |
| Reviewer | 0.05*** | 52.63*** | 4.55*** | 0.30*** | 4.84*** | 0.15*** |
| Child | 0.14*** | 57.03*** | 4.57*** | 0.31** | 11.29*** | 0.36*** |

## C.3. Financial costs of running perturbation

**Design.** To address practical concerns about the computational cost of DBPA's embedding requirements, we conducted a cost analysis across different scales of perturbation studies. We evaluated three state-of-the-art OpenAI embedding models: text-embedding-3-small (2¢ per million tokens), text-embedding-3-large (13¢ per million tokens), and text-embedding-ada-002 (10¢ per million tokens). Our analysis assumes an average of 200 tokens per perturbation, encompassing both input prompts and generated outputs, and scales from small-scale studies (100 perturbations) to extensive analyses (50,000 perturbations).

**Main results.** The cost analysis demonstrates that embedding-based perturbation studies remain highly affordable even at substantial scales. For typical research applications involving 1,000-5,000 perturbations, costs range from 4¢ to $1.30 depending on the chosen model. Even extensive studies with 50,000 perturbations require only $2.00-$13.00 in embedding costs. The text-embedding-3-small model provides the most economical option while maintaining competitive performance, making it particularly suitable for large-scale perturbation analyses. These findings support the practical viability of DBPA, as embedding costs represent a minimal barrier to adoption compared to the computational expenses of generating model outputs themselves.

Table 5: **Cost analysis for embedding-based perturbation studies across different OpenAI embedding models**. Costs are calculated based on official API pricing as of January 2025. Values show the total cost for processing all tokens required for the specified number of perturbations, assuming 200 tokens per perturbation on average. Dollar amounts are shown in parentheses for costs exceeding 100¢ (1 USD).

| # of Perturbations | Total Tokens | OpenAI Embedding Models | | |
| --- | --- | --- | --- | --- |
| | | **text-embedding-3-small (2¢/1M)** | **text-embedding-3-large (13¢/1M)** | **text-embedding-ada-002 (10¢/1M)** |
| 100 | 200,000 | 0.4¢ | 2.6¢ | 2.0¢ |
| 200 | 400,000 | 0.8¢ | 5.2¢ | 4.0¢ |
| 500 | 1,000,000 | 2.0¢ | 13.0¢ | 10.0¢ |
| 1,000 | 2,000,000 | 4.0¢ | 26.0¢ | 20.0¢ |
| 2,000 | 4,000,000 | 8.0¢ | 52.0¢ | 40.0¢ |
| 5,000 | 10,000,000 | 20.0¢ | 130.0¢($1.30) | 100.0¢($1.00) |
| 10,000 | 20,000,000 | 40.0¢ | 260.0¢($2.60) | 200.0¢($2.00) |
| 20,000 | 40,000,000 | 80.0¢ | 520.0¢($5.20) | 400.0¢($4.00) |
| 50,000 | 100,000,000 | 200.0¢($2.00) | 1,300.0¢($13.00) | 1,000.0¢($10.00) |

## C.4. Text similarity alternatives

**Experimental Design.** We investigated whether traditional NLP metrics such as BLEU and ROUGE could be incorporated into the DBPA framework as alternatives to embedding-based similarities. We replicated our role-play experiments using BLEU scores and three ROUGE variants (ROUGE-1, ROUGE-2, and ROUGE-L) as distance measures, computing effect sizes across the same eight personas used in our main experiments. These metrics operate directly in text space rather than semantic embedding space, providing a very different perspective on distributional shifts.

**Main results.** The analysis reveals that traditional NLP metrics can technically function within the DBPA framework, though with important limitations. BLEU scores show particularly high sensitivity to persona-induced changes, with creative personas like 'comedia' and 'child' achieving effect sizes of 0.65-0.66, substantially higher than corresponding embedding-based measures. ROUGE metrics demonstrate more moderate but still significant effects, particularly for creative personas. However, we caution against using such metrics as they primarily rely on text-based tokens.

Table 6: Effect-size estimates using traditional NLP metrics (BLEU and ROUGE) as distance measures within the DBPA framework for role-play experiments. Stars denote significance levels: $^*p < 0.05$, $^{**}p < 0.01$, $^{***}p < 0.001$. While these metrics can technically replace embedding similarities in our framework, they are highly sensitive to small perturbations and may not capture the null distribution effectively. Medical personas (top) generally show lower effect sizes, while creative personas (bottom) demonstrate substantial effects, particularly for BLEU scores. These results should be interpreted cautiously as BLEU/ROUGE operate in text space rather than semantic space.

| | Traditional NLP Metrics | | | |
| --- | --- | --- | --- | --- |
| **Persona** | **BLEU** | **ROUGE-1** | **ROUGE-2** | **ROUGE-L** |
| Act as a doctor | 0.19 | 0.16 | 0.18 | 0.26 |
| Act as a nurse | 0.40$^{***}$ | 0.36$^{***}$ | 0.31$^*$ | 0.28 |
| Act as a medical practitioner | 0.20 | 0.27$^*$ | 0.20 | 0.19 |
| Act as a medical supervisor | 0.24 | 0.34 | 0.34$^{**}$ | 0.26 |
| Act as a comedian | 0.65$^{***}$ | 0.37$^{***}$ | 0.38$^{***}$ | 0.26 |
| Act as a robot from the future | 0.30$^{**}$ | 0.26$^*$ | 0.25 | 0.20 |
| Act as a NeurIPS reviewer | 0.26 | 0.22 | 0.21 | 0.24 |
| Act as a child | 0.66$^{***}$ | 0.44$^{***}$ | 0.49$^{***}$ | 0.33$^{***}$ |

## C.5. Impact of embeddings

**Experimental Design.** We evaluated four distinct embedding models: OpenAI's text-embedding-ada-002 (Ada), alongside three alternative models (Jasper, Stella, and Kalm). Each model represents different approaches to semantic representation. We replicated our core persona experiments across all four embedding functions.

**Main results.** The analysis reveals that while absolute effect sizes vary significantly across embedding models, the relative patterns remain quite consistent. Creative personas such as 'Child' and 'Comedian' consistently produce the largest effect sizes across all embedding architectures, while medical personas demonstrate consistently smaller effects regardless of the embedding choice. This stability indicates that DBPA captures genuine semantic shifts rather than artifacts of specific embedding implementations. The variation in absolute values likely reflects differences in each model's sensitivity to contextual changes and semantic granularity. Notably, Ada embeddings show more conservative effect sizes for medical personas but maintain strong discrimination between persona types, consistent with its design as a general-purpose embedding optimized for broad semantic understanding.

Table 7: **Effect-size estimates by persona across different embedding models.** Stars denote two-sided significance levels: $^*p < 0.05$, $^{**}p < 0.01$, $^{***}p < 0.001$. Higher effect sizes indicate greater distributional shifts in model outputs given the persona instruction. Medical personas (top) are expected to show stable, lower effect sizes, while creative/role-playing personas (bottom) should demonstrate higher variability and stronger effects across embedding models. We find that creative personas consistently produce larger effect sizes across all embedding architectures, with the Child persona showing the strongest effects.

| Persona | Embedding Model | | | |
| --- | --- | --- | --- | --- |
| | Ada | Jasper | Stella | Kalm |
| Act as a doctor | 0.21 | $0.07^{***}$ | $0.07^{***}$ | $0.03^{***}$ |
| Act as a nurse | 0.20 | $0.03^{***}$ | $0.03^{***}$ | $0.02^{***}$ |
| Act as a medical practitioner | 0.19 | $0.05^{***}$ | $0.04^{***}$ | $0.02^{**}$ |
| Act as a medical supervisor | 0.17 | $0.09^{***}$ | $0.08^{***}$ | $0.05^{***}$ |
| Act as a comedian | 0.24 | $0.36^{***}$ | $0.31^{***}$ | $0.23^{***}$ |
| Act as a robot from the future | $0.26^*$ | $0.04^{***}$ | $0.04^{***}$ | $0.05^{***}$ |
| Act as a NeurIPS reviewer | $0.28^*$ | $0.14^{***}$ | $0.13^{***}$ | $0.13^{***}$ |
| Act as a child | $0.33^{**}$ | $0.48^{***}$ | $0.41^{***}$ | $0.35^{***}$ |

# D. Extended related work

Table 8 evaluates the primary related work via five criteria that are important in the context of distribution-based perturbation analysis.

| Method | Example Works | (I) | (II) | (III) | (IV) | (V) | Representative Question |
| --- | --- | --- | --- | --- | --- | --- | --- |
| Measuring Unintended Bias | (Borkan et al., 2019; Dixon et al., 2018; Park et al., 2018) | ✓ | ✗ | ✓ | ✓ | ✗ | Does this model have unintended biases in certain subgroups? |
| Counterfactual Fairness | (Garg et al., 2019) | ✗ | ✗ | ✗ | ✓ | ✗ | How would the prediction change if the sensitive attribute were different? |
| Text Summarization | (Bhandari et al., 2020; Zhang et al., 2019; Zhao et al., 2019; Lin, 2004) | ✗ | ✗ | ✗ | ✓ | ✓ | How well is this text summarized? |
| Distribution-based perturbation analysis | This work | ✓ | ✓ | ✓ | ✓ | ✓ | Do the responses change if we change any input in the prompt? If so, how? |

Table 8: **Related metrics of quantifying text-based outputs. Abbreviations: (I)**: Usable on any black-box model; **(II)**: Any perturbation can be applied; **(III)**: Enables statistical inference; **(IV)**: Computes the effect size; **(V)**: Assumption-free. Distribution-based is the only method that can quantify the effect of any perturbation on an outcome by evaluating, in expectation, the entire distribution of possible outputs. This is important, as single-point evaluations do not capture the stochastic nature of large language models.

