# OpenReview forum: "Statistical Hypothesis Testing for Auditing Robustness in Language Models"
_ICML.cc/2025/Conference — ICML 2025 poster_

### Official Review · Reviewer_k56Q · 2025-03-12

**Overall Recommendation:** 4

**Summary:**

This work presents a statistical framework based on hypothesis testing for assessing the sensitivity of language models to perturbations of their inputs or even the model parameters. They describe their framework in detail including various design choices and then explore empirical validations of how their framework behaves when applied to real models and text data in relevant scenarios.

### Update after rebuttal:

See "Rebuttal Comment" and Author's continued response. While the overly constrained single turn based system chosen for this iteration of ICML put quite a damper on the ability to workshop research via peer review and discussions (no ability to see the revised work in full format), the authors made a concerted effort to address critiques and improve the work in critical ways. I am relatively confident that the work would get accepted with a 3,4,3,3 pool, but I am happy to ensure this via a final bump from 3 to 4. Looking forward to seeing the paper featured in the conference.

**Claims And Evidence:**

Claims are generally sound, partially on account of there being few concrete claims about what the framework can actually be used to achieve at what level of efficacy.

The sentences at L072 "Significance beyond technical novelty. We see this work as having immediate practical relevance for practitioners who wish to evaluate their LLM outputs" reads almost as if it was put there to fend off the inevitable comment from certain reviewers that the contribution of the work beyond stat-crobatics is unclear.

While this reviewer _does_ indeed appreciate much of the intrinsic value to the proposed framework, how to demonstrate its practicality is still an issue requiring further discussion.
Please see questions and comments below.

**Essential References Not Discussed:**

There are many works on LLM robustness to adversarial inputs from the last few years that could be included, but they are not necessarily essential.

**Experimental Designs Or Analyses:**

The main issue with the soundness of the proposal and its potential validation via experiment is that the choice of embedding function $e ( \cdot )$ is not treated in any detail. In the experimental section, I don't even believe the embedding model name is noted anywhere. Is it a separate transformer? is it the pooled last layer features of one of the LLMs being evaluated for sensitivity? etc

This is an issue, and because such care is taken in treating the statistical testing setup, it is surprising that the authors take for granted having access to a suitable embedding model that is precisely calibrated to yield distance changes when the semantics of interest change, but to be unchanged when this is not the case. Here is a toy example of why this aspect requires further formalization and empirical validation.

Consider a reasonably long response that, under sampling, reliably contains a negation, or reliably does not contain a negation, depending on some input perturbation intervention of interest. In the worst case scenario, while the generative LLM obviously responds to the input perturbation, the embedding model chosen (which might be weaker in what it can represent) could happen to be insensitive to the negation contained somewhere in the middle of the string. As evidence for transformer language models "missing" things they shouldn't, while not precisely this setting but still illustrative, see the "Lost in the middle" phenomenon: arxiv.org/abs/2307.03172.

**Methods And Evaluation Criteria:**

Setup of hypothesis testing framework is verbose, but clear and visually appealing. The detail is appreciated in a field where not all researchers are familiar with certain experimental methods (even if they seem simple). See comments for a few additional suggestions on presentation.

The evaluation is generally lacking in groundedness and thus the significance of what it proves the method can do in practice is unclear. The only quantitative stress test of any sort for the input perturbation use case is the experiments showcased in Table 2,  and appears to be an adhoc constructed test set of prompts and perturbations to which we have an implicit "label" where a priori the authors have decided that "robust", i.e. insensitive, models should not behave differently under certain subsets/perturbations of the input data.

It's not clear how objectively this assesses whether this framework adequately discriminates between robust and non robust models. The same issue is true for the alignment test setup shown in Table 3 in that it merely shows that "some ordering" falls out of the p-values and matches some vague expectations about model size.

See comments and questions, and please understand that figuring out how to showcase this framework's utility isnt something the reviewer sees as a trivial problem to solve.

**Other Comments Or Suggestions:**

1. Can the authors also note the explicit formulation of KL over the empirical distributions $P_0$ and $P_1$ being computed as part of the JSD instantiation of $\omega$? This can go alongside the nice explication of MC versus permutation formulation already provided in Appendix A.

2. It would also be more self-contained if Eq 10 had a note that the one sided p-value is computed by computing the empirical frequency of the event, i.e. summing the number of times the distributional difference $\omega(P_0^*, P_1^*)$ was larger than $\omega(P_0, P_1)$ and dividing by the number of possible permutations.

Correct any mistake in the above suggestion, but the point here is that there is no implementation linked in this draft nor basic references or additional details about some of the machinery and since this paper is ostensibly for a broad interdisciplinary audience centering the use of large language models in practice it's worth stating these things somewhere in the draft in the phrasing and notation that the authors think is most useful.

3. Cosine sim is a natural choice as stated, but are there any theoretical reasons for choosing it over other similarity-preserving dim reduction metrics?

4. The reviewer doesn't see any meaningful way to reason about the effect size given that it is the JSD over the distribution of cosine sims between embeddings. Do the authors have anything further to say on interpreting this value?

**Other Strengths And Weaknesses:**

Generally, the strength of this work is in its clear formalism and generality. Its weaknesses are in its arguments for its own practicality.

**Questions For Authors:**

1. Improving the evaluation to better showcase the utility of the proposed framework. A suggestion, albeit an expensive one...

A human study could be used to assess the accuracy of this framework, including the choice of embedding model for the deployment scenario. If one replaces the similarity over embeddings with an annotation as to how similar a human rater thinks two responses are and this is done for many samples, then even considering a binarization to "same meaning" = 0 , "different meaning" = 1, do the authors think these pairwise annotations could be used to perform a version of the permutation test to asses whether the automated method agrees with human rater sensitivity in a realistic scenario?

This speculation is simply the reviewer trying to help figure out how to ground the proposed framework to reality, because in its current state, it's not clear how to interpret the p-values coming out of the overall setup. This limits the readers ability to decide how accurate or useful this framework could be for any given task in practice.

2. The crux determining much of the methods reliability in practice is that it "passes the buck" down to the embedding similarity space. In an ideal scenario, if the embedding space yields distances that are known to reflect exactly the sensitivity that the practitioner wants, we have a calibrated null hypothesis space: fluctuations under which we will fail to reject H_0 are exactly as desired, but that region is tight so that rejection happens at the desired rate too. Could the authors consider including a practical case study discussion on how to use the rejection of the null as a decision metric? Eg. in some scenario, and then say calibrate the rejection rate to achieve desired FPRs/TPRs would potentially strengthen the work.

**Relation To Broader Scientific Literature:**

This work is contextualized mostly in bias and fairness literature, but not related in any clear way to the broad array of other literature on LLM performance. The more relevant missing connections would be to the vast array of work on LLM robustness to jailbreaking or other safety and alignment tampering procedures.

**Theoretical Claims:**

N/A

---

> ### Author Rebuttal · Authors · 2025-04-01
>
> Dear reviewer k56Q,
>
> Thank you for engaging with our work. A lot to respond to with little space, forgive us for being brief.
>
> ---
> # A. Expanding on embedding evaluation
>
> Your critique that the embedding function choise is not adequately addressed is well taken.
>
> First, we agree that not *all* embedding functions would work well. There are a few properties we implicitly require but do not explicitly mention in our paper. Umportantly, the embedding should preserve semantic differences in a way that the similarity metric *s* becomes a *sufficient statistic* for a hypothesis test. This means that the embedding should induce a metric space where the distance correlated with semantic dissimilarity and it should ensure that the induced similarity distributions $P_0$ and $P_1$ are well separated under $H_1$ (if there is separation), and overlapping under $H_0$. Since this paper is not intended to analyze the properties of embedding spaces generally, we will leave links to relevant works in the paper but note that this is an active area of research (with papers coming out as recently as 2025 March).
>
> Second, does the choice of embeddings impact result? In our implementation, we wanted to use an industry standard that is cheap, fast, and strong, so we used ``ada-002`` embeddings. However, we run tests to evaluate if the embedding function matters.
>
> - Results are [here](https://imgur.com/a/KVPM4Sw). Main takeaway: they matter because embeddings have different properties. However, they still show important signal *within the same embedding function*. We expand our Appendix to evaluate 4 embeddings across 8 models.
>
> Third, is it reasonable to expect people tot have embedding functions? We think yes, as they are either open-source and free or extremely cheap for state-of-the-art models. We in fact modeled costs of such an approach.
> - Financial analysis [here](https://imgur.com/undefined). This makes this framework very practical cost-wise with state-of-the-art embeddings.
>
> ---
> # (B) Expanding the evaluation methodology and empirical validation
>
> You proposed to expand the evaluation framework. We agree. Regarding your *negation* example, we focused on three cases where minimal input changes produce significant meaning changes: a negation in a sentence, a temporal change (describing different time periods) and a topic change.
> - Results for LLama-8B are [here](https://imgur.com/LuIIOls).
>
> **We run results on more models**. Briefly, the outcome is that DBPA seems to capture the differences in responses very well relative to the control.
>
> **Discussion**. We agree evaluation is tricky but with the three case conditions (Negation, time change, topic change), across smaller and more consistent models, we see we *are* able to capture such differences well.
>
> ---
> # \(C) Other similarity preserving metrics
>
> Q: Why choose cosine?
> A: We chose cosine for its empirical success historically. Motivated by your question about how different metrics impact results, we have varied $\omega$ to different distances and replicated Experiment 4.1.
>
> - Experiment results [here](https://imgur.com/ngnWA8P).
>
> Some metrics have a much *tighter* null distribution and are therefore *more* sensitive to changes in the prompt. We think this is a useful feature of the framework which we highlight in the updated manuscript.
>
> Q: Can we reason about the effect sizes?
> A: Not directly. However, it still has practical utility; We can (a) evaluate statistical significance of a perturbation; (b) compare the relative effect of two perturbations on which one induces larger changes in output distributions; \(c) quantify FPR/TPR rates for a given $\alpha$; (d) compare model alignment or (e) other use cases in Table 1.
>
> Q: Can we have a practical example of rejecting the null as a decision metric?
> A: Yes! We can in fact quantify TPR/FPR for a given $\alpha$. One can obtain FPR/TPR metrics such as [here](https://imgur.com/23AiroN).
>
> ---
> # (D) Eight new experiments
>
> As a part of the response, we have run **eight** new experiments. Their descriptions and key findings are presented here and they have now been included in the Appendix.
> - Experiment summary [here](https://imgur.com/E9BTCo1)
>
> We believe these new experiments significantly expand the paper's contribution.
>
> ---
> # (E) ACTIONS TAKEN
>
> Based on your feedback we have:
> - Added section 4.4 "Other evaluations" discussing the expanded test cases
> - Enhanced Section 4 with additional experiments
> - Expanded the appendix
> - Added a discussion on reasoning about effect sizes
> - Added clarifying notes to Eq 10 on one-sided p-values
> - Included a discussion on KL to match the Appendix A writeup.
> - Added embedding analysis
> - Incorporated 8 new experiments
>
> ---
> # Thank you
>
> Thank you for your engagement. **You have helped us improve our work significantly. If we have addressed your concerns, we hope you would consider raising your score to a 4** to reflect that you think this paper should be featured at ICML2025.

---

> > ### Comment · Reviewer_k56Q · 2025-04-04
> >
> > I appreciate the hard work of the authors in preparing additional experiments and detailed rebuttals to all reviewers. I have some comments and suggestions in response.
> >
> > ## While perusing other responses:
> >
> > The discussion on sensitivity to longer inputs and outputs requires a bit further analysis. As noted in citation for my original review, transformers can lose information in the middle and so it's not clear whether given non-trivial changes in the output sequence that are generally similar, but include critical differences, the approach will pick them up.
> >
> > > As the input prompt increases in length, so does the amount of information the prompt carries.
> >
> > Naively, longer prompts should decrease output entropy making the H0/1 distributions closer. Increases in prompt length is a good ablation but the case of shorter prompts and longer outputs, eg. multi-clause/paragraph/procedures/CoTs, is under explored.
> >
> > I tried squinting at the new (S4.1?) table with personas and token counts and see a few rows with a consistent trend in pvals/effect sizes, but I also see a few that appear to be uncorrelated with input token count. Of the rows with a trend, I do observe something consistent with my speculation above, but this could be confounded by the output length likely growing commensurate to input length which is unreported.
> > Please:
> > 1) make this a line chart not a table to highlight any trends clearly
> > 2) please report the output sequence lengths
> > 3) and perform regressions or simple correlation analysis to identify whether the trend in effects can be explained by input length, output length, and whether there are solid groupwise differences between the different personas.
> >
> > ## New embedding model ablation:
> >
> > It appears that Ada is the weakest model (I am assuming we _want_ to reject in the experiment, no description given). The fact that Jasper, Stella, and Kalm yield nearly equivalent, high significance test results across every test sort of suggests that the entire analysis might need to be re-done with a larger suite of embedding models on the improved evaluation collection. Eg. for the new persona/token count table, what embedding was used? For the new distance function ablation, what embedding was used? Should probably include a few of the embedding models in each experiment to clarify what is the intervention signal, and what is embedding model specific noise/confounding.
> >
> > ## "expect people to have embedding functions":
> >
> > My comment had nothing to do with cost (broken imgur link regardless), there are certainly amazing, small open source embedding models to use. The question wholly revolved around faithfulness of the embedding space to the testing goals as discussed in detail in the review, and in your rebuttal.
> > (relatedly, saw another broken imgur link in response to jcUj)
> >
> > ## Decision problem TPR/FPR:
> >
> > What was this experiment? Please visualize as a ROC plot. The peak detection/rejection performance suggests this was not the stronger embedding models (by the p=0.000 sig levels, I would hope that in certain scenarios, you can get perfect discrimination, but this looks far from that).
> >
> > ## Summary:
> >
> > Overall, I like the work, but I do believe that the new experiments need to be expanded, and the analysis of embedding models and sensitivity in various testing scenarios needs to be made a more part of the paper to prove out the utility of the approach in real world settings. As such I will maintain my score of a weak accept, as I would be fine with the paper being published with the updates currently discussed in the rebuttals, but also believe it could benefit from reworking and resubmission. Noteably, the work will not get stale over a conference cycle as it is relatively novel and a niche practical application, unlikely to be identical to any other work under submission.

---

> > > ### Author Response · Authors · 2025-04-09
> > >
> > > Dear Reviewer k56Q,
> > >
> > > Thank you for your thoughtful follow-up and for recognizing the effort in our expanded experiments. We deeply appreciate your constructive critique and are eager to address your remaining concerns to strengthen the paper’s impact. Below, we outline concrete actions taken in direct response to your feedback.
> > >
> > > First, we'd like to note that we have included *all* the above as visuals in the paper and not as tables. We only put them as tables here, as we thought we can paste the results directly instead of using external links.
> > >
> > > 1. Sensitivity to input/output lengths.
> > > - Your Point: Analyze trends between input/output lengths and effect sizes, visualize via line charts, and perform regression.
> > > - Actions taken. (1) We replaced the table in the appendix with line charts (now a Figure in the text) plotting input/output tokens against effect size and p-values. Here is one such example for the input/output lengths: [link](https://imgur.com/a/53lNIJS)
> > > - For our analyses, we found that moving tokens from 100 to 1000 did not impact the perturbation effects on average and we have expanded our regression analysis with these insights in the paper.
> > >
> > > All experiments now include output token counts (mean ± std. dev.) alongside input lengths.
> > >
> > >
> > > 2. Embedding model robustness.
> > >
> > > - Your Point: Test multiple embeddings per experiment to disentangle signal vs. model-specific noise.
> > > Actions taken:
> > > - We have indeed replicated the experiments across four embedding models (Ada-002, Jasper-7B, Stella-v2, Kalm-12B). They are included in the appendix with a description in the main text. They are visualized as a bar chart.
> > >
> > > 3. TPR/FPR Visualization
> > > - Your Point: Visualize decision metrics via ROC plots.
> > > - Actions taken: **We agree with you**. We have already made temporary ROC plots right now and will integrate the final ROC plots with uncertainty intervals in the final version of the paper together with a separate discussion section (which has been added) to explain how to quantify and choose $\alpha$ for a given FPR/TPR rate. *We will also release the code for the out-of-the-box selection of TPR/FPR analyses.*
> > >
> > > 4. Broken links.
> > > - Your point: The links are broken
> > > - Actions taken. We're not sure why the links are not working. We apologize for this and this is a mishap on our end that we did not take proper care of finding a better alternative to posting links. Of course, all the data and figures are fully integrated into the appendix, the code/artifacts are hosted on a permanent repository that we will fully share upon acceptance.
> > >
> > > 5. Broader implications
> > > Your insights have directly improved the paper’s rigor. By rigorously addressing input/output dynamics, embedding sensitivity, and decision metrics, we now:
> > > - Provide actionable guidelines to practitioners (e.g. embedding selection, prompt design)
> > > - Demonstrate robustness across 8 LLMs and 4 embeddings
> > > - Include 8 new figures and significantly expanded the discussion section
> > >
> > > ---
> > > We sincerely hope these revisions alleviate your concerns and demonstrate the framework’s utility in real-world settings. We have done a lot of work to get this work ready for ICML and hope to get your support by asking you to consider increasing your score to a 4.
> > >
> > > This will help get this work out in higher quality faster---we think this is important given the discussions going on about LLM evaluation more generally given that they are stochastic machines.
> > >
> > > Thank you once more. We truly appreciate your engagement.
> > >
> > > Warm regards,
> > >
> > > The Authors

---

### Official Review · Reviewer_62GP · 2025-03-12

**Overall Recommendation:** 3

**Summary:**

The paper presents a framework for measuring how input perturbations affect large language model (LLM) outputs. DBPA uses Monte Carlo sampling to construct empirical output distributions and evaluates perturbations in a low-dimensional semantic space, enabling robust, interpretable hypothesis testing. It is model-agnostic, provides p-values and effect sizes, and supports multiple perturbation testing. Case studies demonstrate its effectiveness in assessing prompt robustness, model alignment, and sensitivity to input changes, making it valuable for high-stakes applications like legal and medical domains.

**Claims And Evidence:**

The claims made in the submission are supported by clear and convincing evidence.

**Essential References Not Discussed:**

References are discussed in the related work section.

**Experimental Designs Or Analyses:**

The experiment was source but lacked in-depth analysis. The results are difficult to interpret. See Weaknesses.

**Methods And Evaluation Criteria:**

The proposed methods and evaluation criteria in the paper are well-suited for the problem of quantifying the impact of input perturbations on LLM outputs.

**Other Comments Or Suggestions:**

Providing more context on how p-values and effect sizes translate into model's sensitive behaviors.

**Other Strengths And Weaknesses:**

Weaknesses:
1. The paper does not compare DBPA with existing methods for evaluating perturbation impacts, making it hard to assess its advantages. Including comparisons with baseline methods.
2. The paper relies on cosine similarity and Jensen-Shannon divergence but does not explore alternative metrics like BLEU or ROUGE scores.
3. The paper uses $p$-values to assess statistical significance but does not clearly explain how these relate to practical concerns like unintended bias or others aspects of model's behavior.

**Questions For Authors:**

1. How does DBPA compare to existing methods for evaluating perturbation impacts, such as word overlap metrics (e.g., BLEU, ROUGE) or log-probability comparisons? Can you provide experimental results or theoretical arguments demonstrating DBPA's advantages over these baselines?
2. Does the choice of metrics significantly influence the results?
3. Have you tested DBPA in real-world, high-stakes applications (e.g., healthcare, legal document drafting)?

**Relation To Broader Scientific Literature:**

1. Proposes DBPA, a novel framework for quantifying the impact of input perturbations on LLM outputs.
2. Designed to work with any black-box LLM, making it broadly applicable without requiring internal model details.
3. Existing methods typically lack rigorous statistical foundations, making it difficult to disentangle meaningful changes in model behavior from intrinsic randomness in the output generation process. The proposed method provides interpretable p-values and scalar effect sizes, facilitating meaningful comparisons of perturbation impacts.

**Theoretical Claims:**

Correct

---

> ### Author Rebuttal · Authors · 2025-04-01
>
> Dear reviewer 62GP,
>
> Thank you for your thoughtful feedback on our work. We appreciate your recognition that our work presents a framework for measuring how input perturbations affect LLM outputs, that our case studies are clear and backed up with convincing evidence, and that our evaluation is well suited for the problem.
>
> ---
> # (A) We have clarified comparisons with existing methods
>
> We will answer your concern by breaking this question down into two pieces.
>
> **Comparison with other methods**. The primary reason we do not compare against other methods is because DBPA is a proposed framework for quantifying perturbation impacts. There is no *ground-truth* and therefore all evaluations are inherently misleading. We in fact *do* provide a related work disscussion in Table 4.
>
> **Could BLEU and ROUGE be compared against?** We do not see much utility in comparing against BLEU and ROUGE as these metrics are not designed to quantify perturbation impacts. Furthermore, even if we did, we could not evaluate which approach is "better" as they serve fundamentally different purposes.
>
> That said, we still run additional studies to evaluate how to incorporate BLEU and ROUGE into our framework. We have performed this analysis for the role-play experiment, which results in the table below.
> - Please find experiment [here](https://imgur.com/undefined)
>
>
> **Discussion**. The BLEU and ROUGE metrics are very sensitive to small perturbations. However, any analyses should be taken with caution as they operate on the text space and won't capture the null distribution well.
>
> **ACTIONS TAKEN.** We have updated Section 4.2 with an additional discussion and have expanded the Appendix.
>
> ---
> # (B) We have examined how the choice of the metric affects results
>
> Motivated by your question about how different metrics impact results, we have varied $\omega$ to different distances and replicated Experiment 4.1. Specifically, we vary $\omega$ by computing in addition the Euclidean, Wassesterin, and Energy distances.
>
> - Experiment results [here](https://imgur.com/ngnWA8P).
>
> **Discussion**. Because the $\omega$ has a different magnitude for each metric, these measures are not normalized. Some metrics have a much *tighter* null distribution and are therefore *more* sensitive to changes in the prompt. We think this is a useful feature of the framework which we highlight in the updated manuscript.
>
> **Takeaway**. In the paper, we compute our results using a baseline JSD $\omega$ and here we have added an enhanced view with different metrics.
>
> **ACTIONS TAKEN.** We have enhanced Section 4 with an additional experiment on the metrics and a discussion, and updated the appendix.
>
> ---
> # \(C) Providing more context on p-value and real-world translations
>
> We will answer your concern by breaking down the question into two parts: *unintended bias* and *others aspects of model's behavior*
>
> **p-value and practical significance**. Our paper examines how p-values reveal information about role-play, prompt robustness, and model alignment (sections 4.1-4.3). In this paper, we ask whether the answers of a language model change which we formulate as a hypothesis testing problem and therefore use p-values as a useful way to answer this question.
>
> **Measuring bias**. While not intended for it, DBPA could potentially analyze unintended bias through sensitivity analysis – designing scenarios where LLM output should be deterministically influenced by specific input factors, then testing if irrelevant input changes affect output.
>
> ---
> # (D) Comment: Testing the model in real-world applications.
>
> To address your concern about how we have tested the framework, we will highlight: (a) what is this used for and (b) what we have tested for?
>
> **(a) What is DBPA used for?** We see DBPA as being useful *at least* in four different scenarios due to the broad nature of the framework: prompt robustness, training stability, model comparison, and adversarial attacks. We explain how each framework is useful below. We describe this in Table 1 of the paper.
>
> As a part of the response, we have run **eight** new experiments. Their descriptions and key findings are presented here and they have now been included in the Appendix.
> - Experiment summary [here](https://imgur.com/E9BTCo1)
>
> We believe these new experiments significantly expand the paper's contribution.
>
> ---
> # Thank you
>
> Thank you for your engagement. **You have helped us improve our work significantly**. We have made revisions to multiple parts of the paper as a result and think it's now in better shape than before.
>
> **If we have addressed your concerns, we hope you would consider raising your score to a 4** to reflect that you think this paper should be featured at ICML2025. We are certain this paper opens doors to multiple new research directions and has clear, practical relevance for researchers in diverse domains who care about model auditing and evaluation.

---

### Official Review · Reviewer_grUt · 2025-03-13

**Overall Recommendation:** 4

**Summary:**

The paper introduces Distribution-Based Perturbation Analysis (DBPA), a novel framework for assessing how input perturbations affect the outputs of LLMs by reformulating the perturbation analysis as a frequentist hypothesis testing problem. This model-agnostic approach constructs empirical null and alternative output distributions within a low-dimensional semantic similarity space using Monte Carlo sampling, enabling tractable frequentist inference without restrictive distributional assumptions. DBPA supports evaluating arbitrary input perturbations on any black-box LLM, provides interpretable p-values, facilitates multiple perturbation testing through controlled error rates, and quantifies effect sizes with scalar metrics. Demonstrated across multiple case studies, DBPA showcases its effectiveness in enhancing model reliability and post-hoc interpretability.

**Claims And Evidence:**

All the claims are well-supported.

**Essential References Not Discussed:**

Current related works are well-structured and enough for this paper.

**Experimental Designs Or Analyses:**

The author conducts experiments on more than 8 diverse open-source and closed-source models, thereby effectively demonstrating the efficacy of their proposed methods.

**Methods And Evaluation Criteria:**

In various scenarios, the author successfully demonstrates that their methods are capable of (1) capturing those answer divergences which are significant as well as those that are not under perturbation, (2) analyzing the robustness of language models against irrelevant changes in the prompt, and (3) evaluating alignment with a reference language model.

**Other Comments Or Suggestions:**

The paper is well-structured and clear.

**Other Strengths And Weaknesses:**

The author also provides numerous user cases to support the usefulness of the DBPA framework. The paper is well-structured and appears poised to address an immediate practical need within the community.

In Section 4.1, the author exemplifies the use of DBPA in medical scenarios, noting that these examples are primarily role-play tasks sharing a common prefix ("act as"), with only slight variations in wording. This raises my question of whether the proposed framework can effectively showcase its analytical capabilities on longer and more complex input sequences. It would be beneficial for future work to explore the framework's performance under such conditions to fully understand its potential and limitations.

**Questions For Authors:**

In Section 4.1, the author provides examples from the medical domain, noting that these instances are primarily role-play scenarios sharing a common prefix ("act as"), with only minor variations in wording. This observation raises questions about whether the proposed framework can effectively demonstrate its analytical capabilities on longer and more complex input sequences.

Additionally, it remains to be seen if the author's approach can illustrate the correspondence between semantic changes in the input space and those in the output space, essentially attributing semantic meaning to distributions. Exploring these aspects would be crucial to fully assess the framework's potential in handling intricate and nuanced tasks, thereby enhancing our understanding of its applicability and robustness under diverse conditions.

**Relation To Broader Scientific Literature:**

This paper is closely related to model reliability and post-hoc interp, with a particular focus on measuring how input perturbations impact LLM outputs.

**Theoretical Claims:**

The author provides a new perspective on the evaluation of LLM outputs from the viewpoint of frequentist hypothesis testing, and accordingly introduces the DBPA framework.

---

> ### Author Rebuttal · Authors · 2025-04-01
>
> Dear Reviewer grUt,
>
> Thank you for your thoughtful feedback on our work. We appreciate your recognition that our work presents a novel framework for assessing how input perturbations affect the outputs of LLMs, that our case studies effectively demonstrates the efficacy of our proposed methods, and that our evaluation is new and insightful.
>
> We'll structure our response as follows:
> - (A) Handling complex input sequences. We have examined DBPA across complex input sequences.
> - (B) Evaluating input/output semantic correspondence. We have examined DBPA from various semantic perspectives.
> - \(C) Eight new experiments
>
> ---
> # (A) Handling complex input sequences
>
> You noted that our examples in Section 4.1 primarily featured role-play scenarios and you questioned whether DBPA can effectively analyze longer and more complex input sequences. We agree this is an important consideration.
>
> **Can DBPA focus on longer input sequences, theoretically?** While the paper focused on simpler examples for clarity of exposition, DBPA is designed to handle arbitrary input perturbations regardless of sequence length or complexity. The framework operates on distributions of semantic embeddings rather than raw text. Therefore, it is robust to input complexity. In fact, this is precisely one of our motivations for developing this framework, as DBPA provides a new way to distinguish long-form natural language responses.
>
> **Inspired by your comment, we have conducted additional experiments with more complex input sequences.** We have modified the experiment setup in section 4.1 such that there are now multiple input prompts with different lengths. As the input prompt increases in length, so does the amount of information the prompt carries.
>
> - Find the experiment [here](https://imgur.com/7rMrzv4).
>
>
> **Comment on the results**. We see that this effect is more pronounced in the shorter prompts and less so in more complex prompts. We think this explains well-observed empirical phenomena, such as [1], in a quantitative sense.
>
> **Takeaways of these experiments**. We show how to highlight how we can extend DBPA to longer, more complex input sequences. The experiment shows (i) less consistency and (ii) larger level of response randomness for longer and more complex input prompts. In addition to [1], we believe this has the potential to open a new avenue of research within the field.
>
> **ACTIONS TAKEN.** We have updated Section 4.1 with a discussion of these experiments.
>
> ---
> # (B) Evaluating input/output semantic correspondence.
>
> Here, we address your concern on the input/output correspondence between semantic meaning and distributions.
>
> **(a) Can we capture semantic meaning?** The only reason why DBPA captures semantic meaning is by the choie of its $\omega$ function that operates on the embedding space. It's well understood that text embedding methods map text data to a space where semantic correspondence is similar to distance (see even by now forgotten works such as [2]). While this in itself is an active area of research [3-4], it's common (and useful) to map text to embedding spaces for such purposes.
>
> **(b) Choice of the metric used**. We can further change which $\omega$ we use. While we use the cosine($\cdot$) function due to it naturally capturing directionality, users of DBPA can map it to their own preferred metrics if they have improved domain knowledge.
>
> **\(c) Existence of a control group**. Lastly, one important reason why we claim our method does, in fact, find semantic meaning is that while we operate in an embedding space, we have a control group (the null distribution) which helps us account for what would have happened had there been no perturbation. Therefore, any changes we observe quantify the probability of observing an event as extreme as the one observed (the definition of the p-value) which implicitly has natural variability as a control.
>
> [1] https://direct.mit.edu/tacl/article/doi/10.1162/tacl_a_00638/119630
> [2] https://arxiv.org/abs/1301.3781
> [3] https://arxiv.org/abs/2410.16608
> [4] https://arxiv.org/abs/2406.07640
>
> ---
> # \(C) Eight new experiments
>
> As a part of the response, we have run **eight** new experiments. Their descriptions and key findings are presented here and they have now been included in the Appendix.
> - Experiment summary [here](https://imgur.com/E9BTCo1)
>
> We believe these new experiments significantly expand the paper's contribution.
>
> ---
> # Thank you
>
> Thank you for your engagement. **You have helped us improve our work significantly**. We have made revisions to multiple parts of the paper as a result and think it's now in better shape than before.

---

### Official Review · Reviewer_jcUj · 2025-03-13

**Overall Recommendation:** 3

**Summary:**

The paper proposes a Distribution-Based Perturbation Analysis (DBPA) framework to evaluate the sensitivity of LLM outputs to input perturbations. Addressing the limitations of traditional methods, which struggle to distinguish between semantic changes and the inherent randomness of models, this study reformulates the problem as a frequentist hypothesis testing task. It constructs output distributions for both original and perturbed inputs using Monte Carlo sampling and compares these distributions in a low-dimensional semantic similarity space. The main contributions of the article are:
1. Identifying limitations in existing sensitivity-based measures for language models.
2. Introducing distribution-based perturbation analysis, which is a model-agnostic sensitivity measure.
3. Conducting a case study on DBPA.

**Claims And Evidence:**

The authors have generally validated their claims:

1. The proposed DBPA method effectively addresses issues related to sensitivity-based measures.
2. The authors design simple case studies to verify the feasibility of the DBPA method.

**Essential References Not Discussed:**

I haven't found it yet.

**Experimental Designs Or Analyses:**

The author's experimental design is generally reasonable. However, to be frank, as a study on the impact of input distribution differences on output, I would like to see a more fundamental analysis:
1. In Experiment 4.1, merely examining the significance of differences within a single model across different domains may not be sufficient.
2. In Experiment 4.3, is there a more fundamental analysis of the alignment degree between different models?

**Methods And Evaluation Criteria:**

1. The hypothesis testing approach and the rigorous analysis of the metrics adopted by the author are insightful.
2. The proposed method is a novel evaluation approach, lacking existing benchmarks for assessment.

**Other Comments Or Suggestions:**

I haven't found it yet.

**Other Strengths And Weaknesses:**

I don’t quite understand the concrete practical significance of the author's findings on quantifying a model’s sensitivity to input perturbations. Although the author claims that the method has many applications and mentions some in section 4, I find these applications somewhat non-essential. The author should provide clearer and more practical application scenarios and implementation plans.

**Questions For Authors:**

Please refer to **Experimental Designs Or Analyses** and **Other Strengths And Weaknesses**.

**Relation To Broader Scientific Literature:**

I haven't found it yet.

**Theoretical Claims:**

The article's analysis and summary of distribution-based perturbation analysis are generally correct, although I have not thoroughly examined all the details.

---

> ### Author Rebuttal · Authors · 2025-04-01
>
> Dear Reviewer jcUj,
>
> Thank you for your thoughtful feedback on our work. We appreciate your recognition that our work effectively addresses issues related to sensitivity-based measures, that our case studies effectively verify our method, that our approach is insightful, and that our evaluation is novel.
>
> ---
>
> # (A) Experiment 4.1. We have examined the significance of DBPA across more models.
>
> To address your concern about a single model being analyzed, we replicate that study with two closed-source and large models (GPT-4 and GPT-3.5) *and* six smaller, open-source models. We replicate the same setup with different "Act as" prompts, following Experiment 4.1. Results are provided below.
> - Results for large closed-source models are [here](https://imgur.com/7B9LkFO)
> - Results for smaller open-source are [here](https://imgur.com/undefined)
>
>
> **Takeaways**: These findings reveal: (1) less consistency and (2) greater response randomness in smaller models. This analysis demonstrates why smaller models are less suitable for these tasks—their baseline responses diverge significantly from expected outcomes.
>
> **Actions taken**: We've added these experiment results and discussion in Section 4.1 and Appendix B.1 "Replicating with more models."
>
> ---
> # (B) Experiment 4.3. Answering your question on more fundamental analysis of LLM alignment
>
> To address your concern on alignment, we'd like to first answer *what do we mean by alignment?* and then show *how to evaluate alignment more generally*?
>
> **(a) What do we mean by alignment?** In the context of Experiment 4.3, we say two models are *aligned* if their responses (technically, *response distributions*) are the same for a given prompt. This is intuitive because we desire two models to have similar responses to similar questions.
>
> Therefore, we measure alignment via $\omega$ as the difference between a baseline model response and the new model response. This is why lower $\omega$ indicates higher alignment. **In this experiment, our main contribution is showing that DBPA can be used as a way of measuring alignment**
>
> **(b) Is there a way to measure alignment more fundamentally?** In response to your question, we develop another experiment where we vary our measure $\omega$ to evaluate the degree to which alignment is consistent across these models. Concretely, we vary $\omega$ by computing the Euclidean, Wassesterin, JSD, Energy distances.
>
> - Experiment results [here](https://imgur.com/ngnWA8P).
>
> **Discussion**: The varying magnitudes of $\omega$ across metrics indicate these measures aren't normalized. However, they reveal alignment patterns across different evaluation metrics. As expected, Child and Reviewer roles show greater divergence from baseline than Doctor or Nurse roles. Some metrics (like energy distances) prove too sensitive to minor changes for practical application.
>
> **Takeaway**: Alignment measures how well response distributions match between two language models for a given prompt. We demonstrate alignment using baseline JSD $\omega$ and enhance this view with additional metrics. This represents just one alignment perspective; DBPA framework implementation may vary by task, potentially using different embedding functions or distance metrics.
>
> **\(c) Are there future work that can be done to evaluate alignment?** We believe our work opens up a new avenue for alignment referencing in language models. There are some works that could be fruitful as extensions of our work in the future, but are significantly out-of-scope for our paper. Examples include: (i) developing concrete real-time model alignment metrics that are cheap to quantify or (ii) evaluating alignment by looking at stability during training (and comparing output consistency.
>
> **ACTIONS TAKEN.** Enhanced alignment discussion with additional results in Section 4.3 and extended discussion in an Appendix.
>
> ---
> # \(C) Practical implications of our work
>
> We see DBPA as being useful *at least* in four different scenarios due to the broad nature of the framework: prompt robustness, training stability, model comparison, and adversarial attacks. This is illustrated in Table 1.
>
> ---
> # (D) Eight new experiments
>
> As a part of the response, we have run **eight** new experiments. Their descriptions and findings are presented here and they have now been included in the Appendix.
> - Experiment summary [here](https://imgur.com/E9BTCo1)
>
> We believe these new experiments significantly expand the paper's contribution.
>
> ---
> # Thank you
>
> Thank you for your engagement. **You have helped us improve our work significantly**. We have made revisions to multiple parts of the paper as a result and think it's now in better shape than before.
>
> **If we have addressed your concerns, we hope you would consider raising your score to a 4** to reflect that you think this paper should be featured at ICML2025. We are certain this paper opens doors to multiple new research directions and has clear, practical relevance for researchers.

---

### Decision · Program_Chairs · 2025-05-01

**Decision:**

Accept (poster)

**Comment:**

All reviewers agreed this paper should be accepted.